# Adsorbate-induced formation of a surface-polarity-driven nonperiodic superstructure
Chi Ming Yim [1,2] ✉, Yu Zheng[1], Olivia R. Armitage [2], Dibyashree Chakraborti[2,3], Craig J. Wells[2], Seunghyun Khim [3], Andrew P. Mackenzie [2,3] & Peter Wahl [2,4]

The chemical and electronic properties of surfaces and interfaces are important for many technologically relevant processes, be it in information processing, where interfacial electronic properties are crucial for device performance, or in catalytic processes, which depend on the types and densities of active nucleation sites for chemical reactions. Quasi-periodic and nonperiodic crystalline surfaces offer new opportunities because of their inherent inhomogeneity, resulting in localisation and properties vastly different from those of surfaces described by conventional Bravais lattices. Here, we demonstrate the formation of a nonperiodic tiling structure on the surface of the frustrated antiferromagnet $PdCrO_2$ due to hydrogen adsorption. The tiling structure exhibits no long-range periodicity but comprises few-atom hexagonally packed domains covering large terraces. Measurement of the local density of states by tunnelling spectroscopy reveals adsorption-driven modifications to the quasi-2D electronic structure of the surface layer, showing exciting opportunities arising from electron localisation.

Surfaces and interfaces play a crucial role in modern technology: from information processing, which occurs largely in semiconductor hetero-structures, to large-scale chemical processes facilitated by catalysts. The physics and chemistry of surfaces are as rich as their importance for technological processes: broken symmetry at the surface of a single crystal on its own already leads to novel electronic states and often complex structural reconstructions, providing an opportunity to stabilise entirely different properties at the surface compared to the bulk.

The formation of a surface and concomitant breakage of bonds can result in a range of surface modifications, from mere relaxation, where the surface retains the same symmetries in the directions parallel to the surface as the bulk, to reconstructions, which reduce the symmetry but usually are commensurate with the bulk. Here, we report on the discovery of a new type of nonperiodic tiling structure of hydrogen whose formation is driven by surface polarity in the surface layer from a high-quality single crystal. Due to its lack of periodicity, this tiling structure cannot be described as a commensurate or incommensurate superstructure nor as a quasi-crystalline surface layer. The observed tiling structure self-assembles with structural elements on length scales of a few nanometres, comparable to those of supra-molecular networks[1] however without the need of organic molecules. So far, quasi-crystalline, quasiperiodic or amorphous surfaces have been

reported on quasi-crystalline materials[2] or in thin layers grown on single crystals[3–7] or in 2D polymer networks[8,9].

Nonperiodic tilings and quasi-crystals have exciting consequences for the electronic structure[10,11] and potentially for the catalytic, chemical, and mechanical properties[12]. Moiré superstructures provide comparable opportunities for local variations in reactivity[13]. The significantly more inhomogeneous structure of nonperiodic surfaces compared to that of a normal crystal creates opportunities for tailored sites of different catalytic activities or templates for nucleation of molecules in a range of geometries. Potential applications of these nonperiodic surfaces can go as far as nonstick surfaces for frying pans[14].

Here, we study the surface-polarity driven formation of a nonperiodic tiling structure of hydrogen atoms on the Pd-terminated surface of dela-fossite oxide $PdCrO_2$ (see Fig. 1, a and b side-view crystal structures of $PdCrO_2$ before and after hydrogen adsorption). In the bulk, $PdCrO_2$ is a frustrated magnetic metal[15], with highly anisotropic electronic properties[16,17]. The interplay between the Mott insulating $CrO_2$ layers and the metallic Pd layers that host itinerant electronic states makes $PdCrO_2$ an ideal playground for testing theories of correlated electrons in solids with spectroscopic[18–21] and transport properties[22,23] and provides exciting opportunities to tailor these through thin-film growth[24]. The surfaces of

[1]Tsung Dao Lee Institute & School of Physics and Astronomy, Shanghai Jiao Tong University, Shanghai, China. [2]SUPA, School of Physics and Astronomy, University of St Andrews, North Haugh, St Andrews, Fife, United Kingdom. [3]Max Planck Institute for Chemical Physics of Solids, Nöthnitzer Straße 40, Dresden, Germany. [4]Physikalisches Institut, Universität Bonn, Bonn, Germany. ✉e-mail: c.m.yim@sjtu.edu.cn

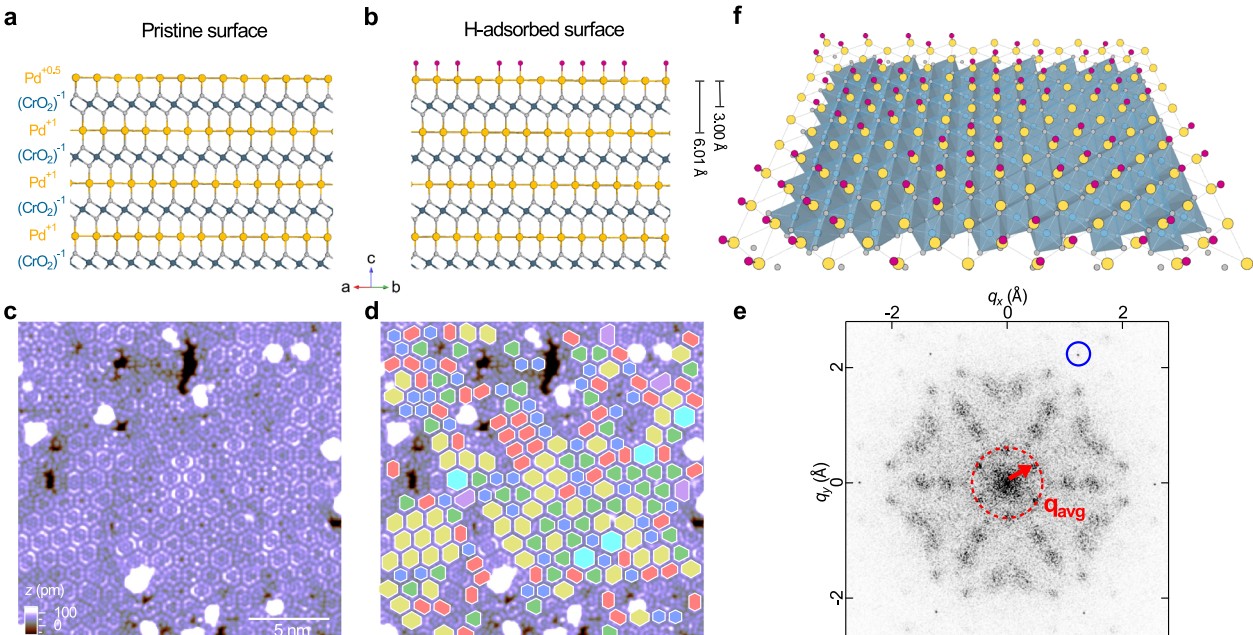

**Fig. 1 | Hydrogen-adsorption induced formation of a nonperiodic tiling phase on the Pd terminated surface of delafossite $PdCrO_2$. a** Side-view crystal structure of delafossite oxide $PdCrO_2$. Inside the bulk, Pd layers possess electrical charge of +1 per Pd atom, layers of $CrO_2$ contain charge of −1 per $CrO_2$ octahedra. Upon cleaving, the vacuum-exposed Pd surface layer possesses charge of +0.5 per Pd atom. **b** As (**a**), following dissociative adsorption of hydrogen, leading to a nonperiodic tiling phase consisting of $(1 \times 1)$-H clusters formed at the surface. **c** Topographic STM image of the Pd-terminated surface of $PdCrO_2$ [$V = -20$ mV, $I = 500$ fA; image size: $(20$ nm$)^2$]. **d** As (**c**), overlaid with an array of hexagons of different colours representing clusters of different types distributed on the surface. **e** Fourier transformation from a $(50$ nm$)^2$ area of the nonperiodic tiling phase. Blue circles mark the Bragg peaks associated with the lattice of Pd atoms. A red arrow indicates the wave-vector, $\mathbf{q_{avg}}$, associated with the spatial arrangement of the hydrogen clusters. The overlaid red dashed circle has a radius ($\sim 0.61$ Å$^{-1}$) equal to $|\mathbf{q_{avg}}|$. **f** Structural model of the Pd surface layer observed in this study. The surface is chemi-sorbed by hydrogen, leading to a tiling phase that lacks any periodicity. In the model, raspberry-red spheres represent atomic H adsorbed on each Pd atom in the surface layer.

many delafossites are polar as a result of the charge distribution within the unit cell. In $PdCrO_2$, the Pd ions carry a nominal charge of +1 and the $CrO_2$ layers of −1 (Fig. 1a)[25]. This means that the surface will be susceptible to electronic or structural reconstruction to avoid a polar catastrophe[26]. Electronic reconstruction is observed for the Pd and $CoO_2$ surface terminations of $PdCoO_2$[27–30] and for the $CrO_2$ termination of $PdCrO_2$[21], although structural rearrangements are also found[31]. Beyond electronic correlation effects, the Pd surface of $PdCoO_2$ has been reported to exhibit excellent electrocatalytic activity for the hydrogen evolution reaction[32,33], surpassing nanostructured palladium[34–36] and has been found highly reactive, showing evidence of adsorption of gas even in ultra-high vacuum[37].

## Results
### STM appearance of the Pd-terminated surface of delafossite $PdCrO_2$

We cleaved high-quality single-crystal samples of delafossite $PdCrO_2$ normal to their $c$-axis at a *nominal* temperature of ~20K. Due to the crystal structure and the nature of chemical bonding, cleavage occurs typically between the Pd and the $CrO_2$ layers, resulting in two predominant surface terminations: a Pd- and a $CrO_2$-terminated surface (details on distinguishing between the two surface terminations in scanning tunnelling microscopy (STM) are provided as Supplementary Notes 1 and 2 and Supplementary Figs. 1 and 2). Here, we concentrate on the Pd surface termination. Figure 1c shows a large-scale topographic image of a Pd-terminated surface, showing that the surface does not appear flat as seen in $PdCoO_2$[38], but is characterised by arrays of hexagonal islands of different sizes, shapes, and orientations tightly packed with each other. As a visual guide, in Fig. 1d we show the same image overlaid with hexagons of different colours representing islands of different types. The Fourier transformation (Fig. 1e) shows clear and sharp peaks due to the atomic lattice, less sharp peaks (indicated by a red arrow) due to the spatial arrangement of the

hexagonal islands, and other broad features originating from the internal atomic arrangements within those islands. The lack of a single motif in this nanostructured phase results in a not so well-defined periodicity in the arrangement of the islands, in turn leading to broadening of the related features in the Fourier transformation. All of these are testament of the nonperiodic nature of the observed tiling structure. The topographic image reveals islands of distinct sizes that we shall henceforth call clusters.

From a detailed comparison with density functional theory (DFT) calculations (see the Methods section), we find that the observed nonperiodic tiling structure can be understood as a result of dissociative adsorption of hydrogen from the residual vacuum, in which each H atom bonds directly on top of one Pd atom, forming a $(1 \times 1)$-H ordered structure within each cluster (see Supplementary Note 3 and Supplementary Figs. 3–5 for the calculation results in full, and Supplementary Fig. 6 for an experimental STM image confirming our assignment of the site occupation of hydrogen within the clusters). Their adsorption behaviour on the Pd-terminated surface of $PdCrO_2$ observed here is very different from that on the {111} surface of Pd single crystals, where H preferably adsorbs at the hollow sites and their adsorption does not lead to any tiling phase as the one observed here[39]. To better illustrate the hydrogen-induced tiling phase observed in STM, we show in Fig. 1f a schematic model of the tiling structure. The surface exhibits a complex order that consists of characteristic groups of atomic hydrogen that form clusters of different sizes and shapes (as represented by the overlaid hexagons of different colours in Fig. 1d), and line segments without hydrogen that form the boundary between the clusters. These patterns are tiled in a space filling manner in two dimensions, but without resulting in any distinct periodicity, as can be seen in the Fourier transformation of the topographic image in Fig. 1e.

We also note that in addition to the $(1 \times 1)$-H clusters, the surface is also populated with a number of defects that are present in the form of lump-and-hole pairs. Formed as a result of surface cleavage, these defects account

for a surface defect concentration of ~5.3%, and therefore have only a minor effect on the overall surface structure (see Supplementary Note 4 and Supplementary Fig. 7).

### Spectroscopic signature of hydrogen and the properties of the clusters

The hydrogen chemisorbed on the Pd surface termination can be detected through its vibrational modes in inelastic electron tunnelling spectroscopy. In Fig. 2a we present a $d^2I/dV^2$ tunnelling spectrum taken at the centre of one of the clusters, showing sharp features symmetric around zero bias at energies $|E| = 42$ meV, 84 meV and 272 meV, respectively. These are clear signatures of inelastic tunnelling. From our DFT calculations (see the Methods section) we obtain vibrational modes of the chemisorbed hydrogen at energies of 61 meV for the in-plane vibrational mode and 257 meV for the out-of-plane mode, in good agreement with the experiment. Similar values were obtained in previous DFT studies of the surface phonon modes of H/Pt(111)[40], where the in-plane vibrational mode was found at 47.4 meV, and the out-of-plane mode at 277.2 meV.

For the $(1 \times 1)$-H tiling structure formed on the Pd-terminated surface, it would be expected that the apparent barrier height, a proxy for the surface work function, exhibits significant variation between Pd atoms with and without adsorbed hydrogen. We show in Fig. 2b, c a topographic $z(\mathbf{r})$ image and a map of the local barrier height $\phi(\mathbf{r})$ taken simultaneously from a small region of the nonperiodic tiling. Also shown is a line cut through the $z(\mathbf{r})$ image and the $\phi(\mathbf{r})$ map across several clusters, see Fig. 2d. The $\phi(\mathbf{r})$ value varies substantially throughout the tiling, and reaches its maxima (~7 eV) and minima (~5 eV) at the centre of the clusters and the boundary region, respectively. Consistent with the calculation results (see Supplementary Note 3 and Supplementary Fig. 3 for details), our $\phi(\mathbf{r})$ data further support the scenario of electron transfer from the Pd surface to the adsorbed hydrogen, followed by an increase in the surface work function.

The charge transfer also results in a Coulomb repulsion between the hydrogen atoms and consequently their displacements towards the boundary. From a high-resolution image, we have analysed the lateral positions of the hydrogen atoms within each of the clusters. Figure 2e shows a close-up image of the nonperiodic tiling with crosses marking the positions of the Pd atoms within the Pd surface layer, with their positions determined by numerical fitting using a 2D Gaussian function to every spot on the Pd lattice appearing in the box-filtered image, which was itself generated using only the lattice Bragg peaks in the Fourier transformation, see Supplementary Fig. 8 for details. It can be seen that several of the hydrogen atoms in the clusters relax away from their ideal positions (see Fig. 2f, g). Through a detailed analysis of the atomic positions, we find that hydrogen atoms at the periphery of the clusters displace more from their ideal positions than those at the centre (Fig. 2h).

We also find that clusters of different sizes and shapes are spatially arranged to have an edge-to-edge separation distance between adjacent clusters of $\sim \sqrt{3}a$, where $a$ is the lattice constant of the Pd surface layer. Having such a separation distance also means that the boundary region separating the clusters is one atom wide, confirming its composition of lines of Pd atoms with no adsorbed hydrogen, in turn explaining the less visible appearance of the boundary region in STM.

### Neural network characterisation and bias dependent appearance of the nonperiodic tiling

To analyse the spatial arrangement of the $(1 \times 1)$-H clusters in the surface layer in more detail, we used a neural network to categorise the clusters seen in the STM images according to their types. We trained the neural network to recognise the different arrangements of atoms as shown in Fig. 3a using YOLOv3 (see the Methods section for technical details) from a training data set consisting of hundreds of images of individual clusters. We designated the different types of clusters by a symbol $T_k$, where $k$ represents the number

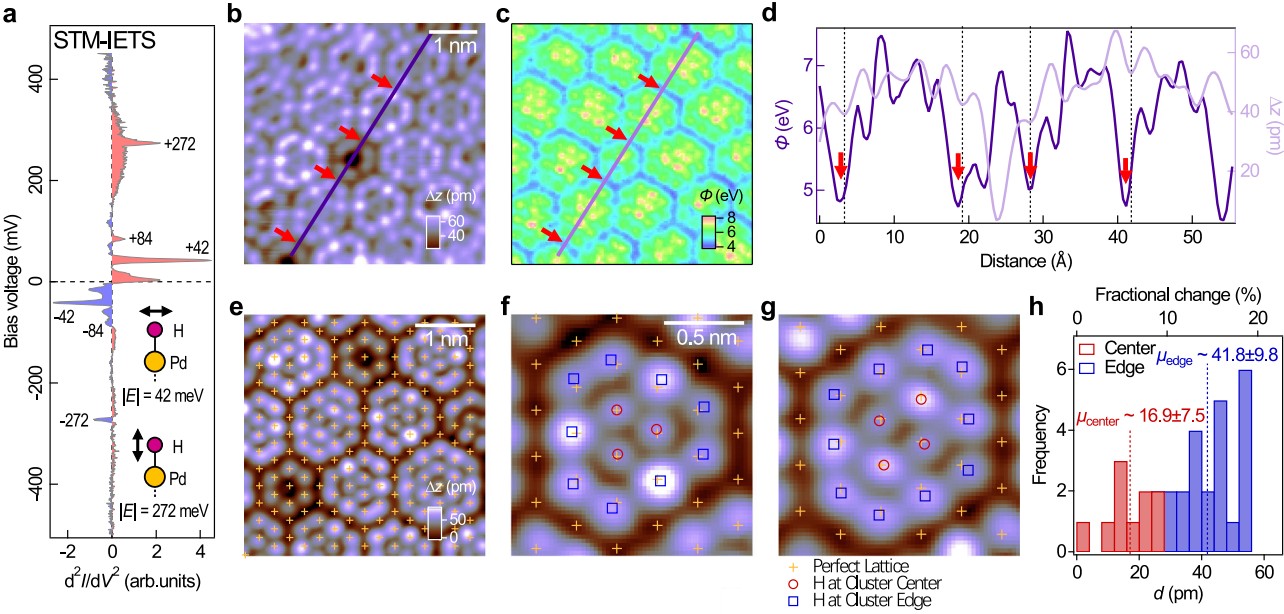

**Fig. 2 | Chemical signatures of the $(1 \times 1)$-H clusters. a** Point inelastic electron tunnelling ($d^2I/dV^2$) spectrum recorded from the central position of a $(1 \times 1)$-H cluster ($V_s = 500$ mV, $I_s = 100$ pA; $V_{mod} = 5$ mV; 10 sweep average). The spectrum shows three sets of inelastic peaks at $|E|$ of 42 meV, 84 meV (which is the second harmonic of the 42 meV peak) and 272 meV respectively. Insets: schematics showing the in-plane and out-of-plane vibrational modes of the H-Pd bond at 42 meV and 272 meV respectively. **b** STM topographic $z(\mathbf{r})$ image of the nonperiodic tiling [$V = 200$ mV, $I = 10$ pA; image size: $(5 \text{ nm})^2$]. **c** Corresponding local barrier height $\phi(\mathbf{r})$ map extracted from the $I(\mathbf{r}, z)$ spectroscopic imaging data recorded simultaneously with (**b**). **d** Topographic (light-) and $\phi$ (dark-purple) line-cuts taken along the same line in (**b**) and (**c**). Arrows indicate the positions at the boundary at which $\phi$ reaches local minima. **e** Another $z(\mathbf{r})$ image of the $(1 \times 1)$-H tiling phase [$V = 32$ mV, $I = 32$ pA; image size: $(4 \text{ nm})^2$]. **f, g** Close-ups of two individual clusters extracted from (**e**) [image size: $(1.5 \text{ nm})^2$]. In (**e**)–(**g**), crosses mark the positions of Pd atoms in the perfect triangular lattice. In (**f**) and (**g**), red circles and blue squares mark the positions of the hydrogen atoms at the centre and at the edges of the $(1 \times 1)$-H clusters respectively. **h**, Histograms of the in-plane displacement of the hydrogen atoms (relative to their beneath Pd atoms) at the centre (red) and edges (blue) of the hydrogen clusters in (**f**) and (**g**), as well as of one other cluster (not shown).

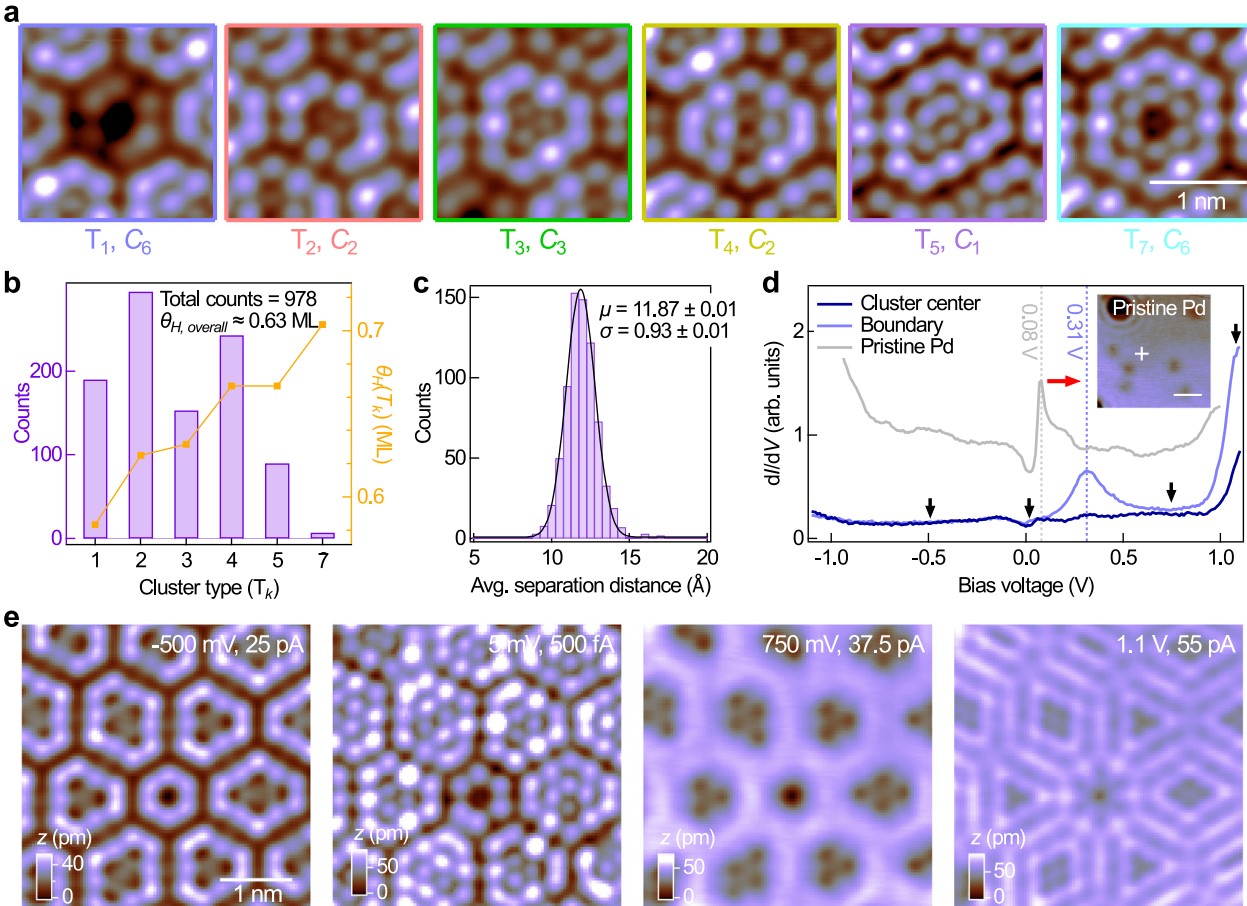

**Fig. 3 | Distribution and spatial arrangement of the hydrogen clusters, and their bias-dependent appearance in STM. a** Zoom-in topographic images of the six most popular $(1 \times 1)$-H cluster types present within the nonperiodic tiling [$V = 50$ mV, $I = 1$ pA; image size: $(2$ nm$)^2$]. Symbols $T_k$ denote different cluster types, where $k$ in the subscript denotes the number of hydrogen atoms present in the central part in each cluster type. The rotational symmetry possessed by each cluster type is also provided. **b** Number distribution of the different cluster types, calculated from a $(50$ nm$)^2$ image (Supplementary Fig. 10) of the nonperiodic tiling using the YOLOv3 neural network approach. Threshold for detection = 0.7. Hydrogen coverage within each cluster type is also plotted, with the overall hydrogen coverage in the $(50$ nm$)^2$ image calculated to be $\approx 0.63$ ML. **c** Histogram of the average separation distance between neighbouring clusters, extracted from their predicted locations within the same image. A Gaussian fit to the histogram yields mean and standard deviation

values of $\mu = 11.87 \pm 0.01$ Å and $\sigma = 0.93 \pm 0.01$ Å respectively. **d** Point $g(V)$ spectra recorded at the centre of one of the hydrogen clusters and at a position within the boundary region ($V_s = -1.25$ V, $I_s = 1$ pA; $V_{mod} = 10$ mV). For comparison, the spectrum recorded from a defect-free position on the pristine Pd-terminated surface of PdCrO$_2$ ($V_s = 1$ V, $I_s = 500$ pA; $V_{mod} = 10$ mV) is also provided and is vertically offset for clarity. A red arrow indicates a shift in energy position of the sharp d$I$/d$V$ peak from 80 to 310 meV before and after adsorption of hydrogen. Inset of (**d**), $(8$ nm$)^2$ STM topographic image taken from the pristine Pd-terminated surface of PdCrO$_2$ ($V = 200$ mV, $I = 50$ pA; scale bar: 2 nm). The cross indicates the position at which the grey spectrum in (**d**) were taken. **e** Bias-dependent topographic images taken at the same position on the nonperiodic tiling [image size: $(4$ nm$)^2$]. The bias voltages at which the images were recorded are indicated by the black arrows in (**d**).

of hydrogen atoms in the central part of the cluster (also see Supplementary Fig. 9 the STM images showing the presence of an adsorbed H atom at the central position of clusters $T_1$ and $T_7$).

Our results (Fig. 3b) from the analysis of a $(50$ nm$)^2$ image (Supplementary Fig. 10) indicate that among the detected clusters, the $T_2$ clusters are the most common, followed by other cluster types, including $T_4$, $T_1$, $T_3$, and $T_5$. The rarest are $T_7$ clusters. Note that the non-observation of any $T_6$ cluster is probably due to its lack of symmetry (rotational or mirror) as compared to other cluster types. In particular, the distribution indicates the prevalence of the two-fold rotationally ($C_2$) symmetric cluster types $T_2$ and $T_4$ with an even number of atoms in the centre. Taking into account the boundary positions shared among adjacent clusters, we calculated the hydrogen coverage [in monolayer (ML)] within each cluster type [$\theta_H(T_k)$, Fig. 3b], and determined the overall coverage of hydrogen in the imaged area to be $\approx 0.63$ ML, seemingly slightly overcompensating the surface polarity. From the results of our analysis, we propose that the number distribution of the clusters is governed by the intricate balance between the hydrogen coverage, repulsive interactions between the adsorbed hydrogen and

possible charge transfer between the adsorbed hydrogen and the Pd surface layer.

Determining the cluster positions also allows for the extraction of their mean distance (Fig. 3c). A Gaussian fit to the histogram reveals a mean distance of $\mu \sim 11.9$ Å with a standard deviation of $\sigma \sim 0.9$ Å. Taking $\mu$ as the separation distance of the clusters arranged in a perfectly close-packed array, we have calculated the associated wavevector to be $|\mathbf{q}| = 0.61 \pm 0.05$ Å$^{-1}$. This calculated value agrees perfectly with that arising from the spatial arrangement of the clusters as shown in the Fourier transformation (Fig. 1e).

Assuming that all clusters have the same size and shape, and using $\mu$ as the lattice constant of the cluster unit cell, we, by comparison of the areas between the cluster and the primitive unit cell, determined that each cluster on average contains $\sim 16.8$ atoms. This is very close to the number of atoms ($=16$) contained within each $T_2$ cluster, in which atoms in the boundary regions that are shared with adjacent clusters are also counted.

The formation of the $(1 \times 1)$-H tiling phase also leads to a strong spatial variation of the local electronic structure across the tiling, as reflected in drastic differences in the differential conductance spectra $g(V)$ acquired in

the centre of a cluster and at a boundary position between the clusters (Fig. 3d), together with their bias-dependent appearance in STM (Fig. 3e), revealing dramatic changes in their apparent height. Furthermore, comparing the $g(V)$ spectra obtained from the nonperiodic tiling with that of the pristine surface (grey curve), we observe that the sharp conductance peak at ~80 meV found on the pristine surface has shifted to ~310 meV after hydrogen adsorption, which we attribute to electron transfer from the surface to hydrogen, as is also seen from the projected density of states from the DFT calculations (Supplementary Fig. 5).

### Electronic confinement in the hydrogen clusters

The nonperiodic nature of the observed tiling and the different cluster sizes create an electronic landscape governed by localisation with spatially varying bound state energies due to quantum confinement. The strong influence of the nonperiodic tiling on the electronic structure is readily visible in the bias-dependent images (Fig. 3e). The local confinement occurs on a smaller energy scale and can be visualised by spectroscopic imaging. In Fig. 4 we show a topographic image (Fig. 4a) and spectroscopic $g(V, \mathbf{r})$ map slices (Fig. 4b–d) recorded from a $(5\,\text{nm})^2$ region of the $(1 \times 1)$-H tiling structure, showing that the bound states associated with each of the different types of clusters exhibit an increase in the density of states at different energies.

To demonstrate this, in Fig. 4e–h, we show the characteristic tunnelling spectra extracted from cluster types $T_1$ to $T_4$. As shown in Fig. 4e, the spectrum extracted at the corner position of a $T_1$ cluster (light blue) exhibits a sharp peak at $E_p = -20$ meV, which is not present in that of the central position (dark blue). This difference is also manifested in the $g(V, \mathbf{r})$ map slice at the peak energy $eV = E_p$ (inset of Fig. 4e), where strong maxima are seen at the corners of the $T_1$ cluster, leading to a shape like a flower with six petals. Similar resonant states are present in other cluster types including $T_2$ and $T_4$. In $T_2$ clusters, the electronic states at $E = -62.5$ meV (Fig. 4f) are localised at the corners near the long edge (Fig. 4f). In $T_4$ clusters, the states at $-10$ meV (Fig. 4h) are localised at the short edges of the cluster, and at the positions near the centre (inset of Fig. 4h). However, we do not observe clear

localised states within $T_3$ clusters (see Fig. 4g and its inset), which we speculate to be related to the fact that unlike $T_1$, $T_2$ and $T_4$ clusters that have at least one pair of parallel, equal length edges within each cluster, $T_3$ clusters do not have any such edge pair. Without such an edge pair, strong electronic resonance cannot occur within the cluster. Our results therefore show clear signatures of quantum confinement within the clusters. The fact that the bound states are seen only for negative bias voltages suggests that they originate from a hole-like band with its band top close to the Fermi energy.

## Discussion

Our results show the adsorption induced formation of a nonperiodic tiling structure consisting of $(1 \times 1)$-H clusters of different types on a perfectly ordered high-purity single-crystal surface. The surfaces of $PdCrO_2$ are polar surfaces - in the bulk adjacent layers of Pd and $CrO_2$ exhibit a nominal valence of $+1$ and $-1$, which leaves the bulk charge neutral, but creates a polar catastrophe at the surface unless lifted by a reconstruction, be it electronic or structural. Previous work shows that the Pd surface terminations of both $PdCoO_2$ and $PdCrO_2$ exhibit ferromagnetic surface states of highly itinerant electrons[28]. Reaction of the surface with molecular hydrogen from residual vacuum results in a nonperiodic tiling superstructure of chemisorbed hydrogen with properties drastically different from the chemically homogeneous Pd layer. Evidence for hydrogen adsorption has previously been observed for the Pd-termination of $PdCoO_2$[37]. This interpretation is supported by measurements of the local density of states and the local barrier height, where we observe large variations between Pd sites with and without adsorbed hydrogen. Our interpretation is also consistent with the significant relaxation of hydrogen within the clusters and the vibrational modes detected within them. Scans performed at high bias voltage can lead to desorption of hydrogen from the tiling structure and concomitant recovery of a pristine surface (see Supplementary Fig. 11). As a result, other possibilities for the formation of the tiling structure, including the formation of Pd vacancies at the boundary (which would result in significant Pd deposits elsewhere on the surface), are naturally ruled out.

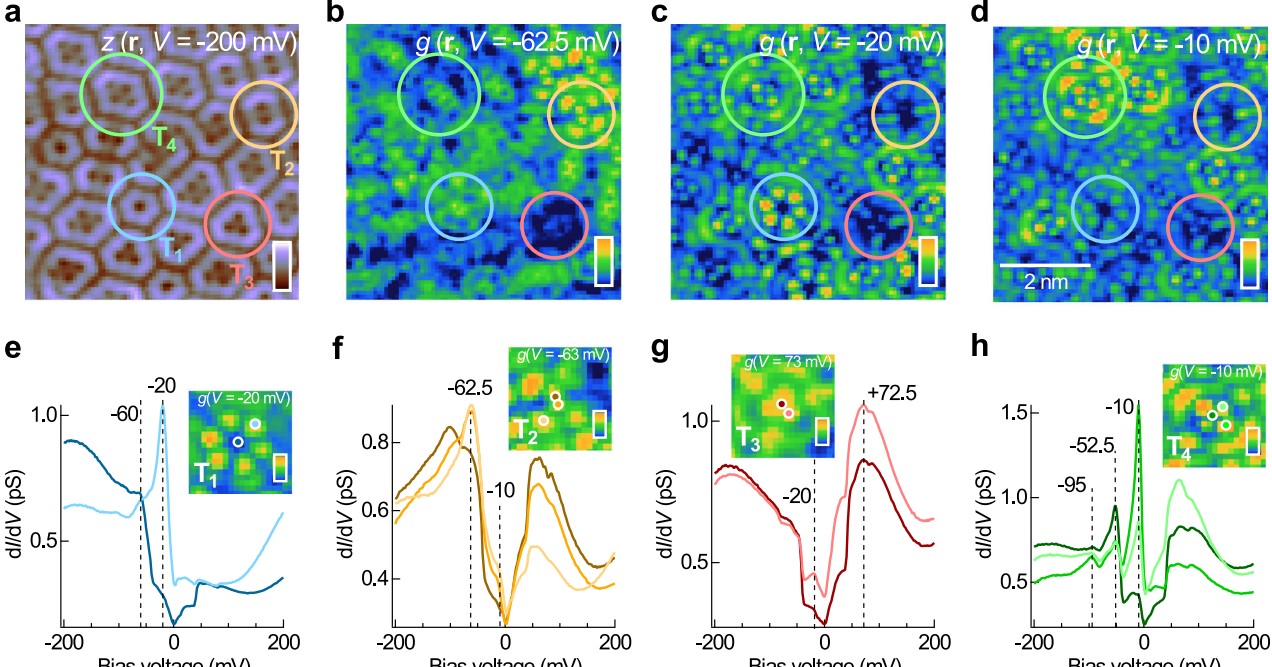

**Fig. 4 | Spatial confinement induced electronic bound states formed within the $(1 \times 1)$-H clusters. a** STM topographic image of the nonperiodic tiling structure $[V = -200$ mV, $I = 600$ fA; image size: $(5\,\text{nm})^2]$. **b–d** $g(\mathbf{r}, V)$ map slices at different energies simultaneously recorded with (**a**): **b** $-62$ meV, **c** $-20$ meV, and **d** $-10$ meV ($V_s = -200$ mV, $I_s = 50$ pA; $V_{\text{mod}} = 2.5$ mV). In (**a**)–(**d**), circles of different colours mark the $(1 \times 1)$-H clusters of different types: $T_1$ to $T_4$. **e–h** Point $g(V)$ spectra extracted from different positions in each of the clusters $T_1$ to $T_4$. Insets of (**e**)–(**h**), Topographic images of the clusters [image size: $(1.5\,\text{nm})^2]$, overlaid with markers indicating the positions at which the $g(V)$ spectra were extracted.

The formation of the tiling structure of hydrogen in the Pd surface layer of $PdCrO_2$ also results in a localisation of the electronic states close to the Fermi energy due to quantum confinement, confirmed through our spectroscopic investigations: the electronic structure varies significantly between different types of clusters, showing strong resonant states in the vicinity of the Fermi energy.

Furthermore, in addition to confirming that the composition of the tiling structure depends on the temperature at which it forms, we also note that the tiling structure is absent from the 12 K-cleaved sample, which itself reveals a metallic surface (see Supplementary Fig. 12) with similar spectroscopic signatures as seen on the Pd-terminated surface of $PdCoO_2$[38]. We attribute the absence of the tiling on the 12 K-cleaved sample to the huge reduction of the hydrogen partial pressure inside the vacuum chamber as a result of cold trapping by the liquid helium-cooled surfaces during sample cleavage, in turn leading to no adsorption of hydrogen on the sample surface.

The electronic and structural inhomogeneity of the adsorption-induced tiling structure on the Pd-termination reveals new opportunities to design surfaces and template substrates with a range of catalytically active sites, with targeted reactivity based on geometry and local electronic structure. An interesting open question is how hydrogen adsorption affects the strongly correlated electronic states in the Mott insulating $CrO_2$ layer. The anisotropy of the electronic transport properties of the material, together with the highly two-dimensional nature and extremely high chemical reactivity of the Pd surface termination may provide important clues to understand the excellent electrocatalytic activity of some delafossites for the hydrogen evolution reaction[32,33] and underpin future rational design of delafossite-based catalysts that take advantage not only of the electronic configuration but also the nonperiodic nature of the surface structure.

## Methods

### Scanning tunnelling microscopy/spectroscopy (STM/S) measurements

The STM/S measurements were performed using a home-built low temperature STM that operates at a base temperature of 1.8 K[41]. Pt/Ir tips were used, and were conditioned by field emission performed on a Au target before use. Differential conductance $dI/dV$ or $g(V)$ and inelastic electron tunnelling $d^2I/dV^2$ spectra, and spectroscopic maps were recorded using a standard lock-in technique, with the frequency of the bias modulation set at 413 Hz. All reported data were obtained at a sample temperature of 4.2 K unless otherwise stated. For STM/S measurements, the clean sample surfaces were prepared by in-situ cleaving at a *nominal* sample temperature of ~20 K.

To examine the reproducibility of the nonperiodic tiling structure, similar STM/S measurements were performed using a commercial *Unisoku* USM1300 ultrahigh vacuum (UHV) STM machine that can operate at a base temperature of 300 mK, where clean sample surfaces were prepared by in-situ cleaving of the samples held at sample temperatures of 78 K or 12 K under the UHV condition (base pressure $7 \times 10^{-11}$ mbar). In the experiments, we observed a similar tiling phase formed on the Pd terminated surface of the sample cleaved at 78 K, but a pristine, unreconstructed Pd surface of the sample cleaved at 12 K.

### Crystal growth

Single-crystal samples of $PdCrO_2$ were grown by the NaCl-flux method as reported in ref. 42. First, polycrystalline $PdCrO_2$ powder was prepared from the following reaction at 960 °C for four days in an evacuated quartz ampoule:

$$2LiCrO_2 + Pd + PdCl_2 \rightarrow 2PdCrO_2 + 2LiCl. \qquad (1)$$

The obtained powder was washed with water and aqua regia to remove LiCl. The polycrystalline $PdCrO_2$ and NaCl were mixed in a molar ratio of 1:10. Then, the mixture in a sealed quartz tube was heated at 900 °C and slowly cooled down to 750 °C. $PdCrO_2$ single crystals were harvested after dissolving the NaCl flux with water.

### Neural network analysis

The YOLOv3 neural network was trained using codes from ref. 43, to identify the six most common cluster types using the images containing individual clusters of each cluster type. The training and testing data sets consisted of eight and five images respectively for each class. To ensure that the network was trained on clusters at different angular orientations, to expand the data set, copies of each image were rotated at 60° intervals. This resulted in a total of 282 images used for training and 174 for testing. The $z$ values in the images were re-scaled to have a zero mean ($\bar{z}$) and unity standard deviation ($\sigma_z$). When applying the neural network to the $(50$ nm$)^2$ image, rescaled $z$ values that exceed $3\sigma_z$ from $\bar{z}$ on both sides were excluded due to the presence of surface defects. The rescaling and exclusion of the extreme values were then repeated to produce the final image. This contrast saturation approach provided better contrast for the surface clusters, in turn allowing for a higher detection rate by the network.

### DFT calculations

To determine the origin of the nanocluster network and valence disproportionation, we performed extensive DFT calculations to model a number of scenarios. Ultimately, the one that best matches the experiments and captures all observations is that the valence change is induced by hydrogen adsorbed on the Pd surface, resulting in a local change of the work function and significant local modification of the electronic structure of the Pd layer through hole doping. The calculations were performed on a two-layer slab of $PdCrO_2$ with a range of unit cell sizes. Scenarios that we investigated include a vacancy network, adsorption of hydrogen and carbon monoxide, and structural reconstructions. For the calculations, we added a Coulomb repulsion $U$ for the $d$-orbitals of the Cr, consistent with previous studies[21], however note that the inclusion of the $U$ term does not result in any significant differences for the surface layer. DFT calculations were performed using PBE functionals using VASP[44–49] with a plane wave energy cut-off of 800 eV. Most calculations were performed for the unit cell of a $T_2$ cluster, with a $2 \times 2$ k-grid. For structural relaxations, the bottom unit of $PdCrO_2$ and the bottom Pd surface were fixed. The bottom surface of the slab was saturated with the same number of hydrogen atoms as the top surface so that both the bottom and top surfaces have the same chemical configuration.

## Data availability

The data that support the findings of this study are openly available at https://cstr.cn/32010.11.sjtu.scidata.00000050.

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

## Acknowledgements

We gratefully acknowledge discussions of hydrogen adsorption with Phil King and of preliminary calculations with Chiara Gattinoni, as well as discussions with Federico Mazzola and Gesa Siemann. We further thank Neville Richardson for critical reading of and valuable comments on the manuscript. C.M.Y. and P.W. acknowledge support from EPSRC (EP/S005005/1). C.M.Y. acknowledges additional support from the Ministry of Science and Technology of China (2022YFA1402702, 2021ZD0302700) and TDLI Start-up Fund. A.P.M. acknowledges support from the MPG. This work used the Cirrus UK National Tier-2 HPC Service at EPCC funded by the University of Edinburgh and EPSRC (EP/P020267/1), as well as the HPC cluster Hypatia of the University of St Andrews.

## Author contributions

C.M.Y., P.W. and A.P.M. conceived the project. C.M.Y., Y.Z. and D.C. performed STM experiments. C.M.Y. analysed the data and prepared the figures. O.R.A. performed neural network analysis. C.J.W. prepared the training data set. S.K. and A.P.M. grew the crystals. P.W. performed DFT calculations. C.M.Y., P.W. and A.P.M. wrote the manuscript. All authors discussed and contributed to the manuscript.

## Competing interests

The authors declare no competing interests.
