## [Transparent Peer Review file · Communications Materials]

Adsorbate-induced formation of a surface-polarity-driven non-periodic superstructure

Corresponding Author: Professor Chi Ming Yim

Version 0:

Decision Letter:

Dear Professor Yim,

Thank you for submitting your manuscript, "Adsorbate-induced formation of a surface-polarity-driven non-periodic superstructure", to Communications Materials. It has now been seen by 2 referees, whose comments are appended below. You will see that while they find your work of great interest, an important point is raised by Reviewer #2 regarding the control over hydrogen adsorption on the surface. We are interested in the possibility of publishing your study in Communications Materials, but would like to consider your response to these concerns in the form of a revised manuscript before we make a decision on publication.

We therefore invite you to revise and resubmit your manuscript, taking into account the points raised.

When submitting your revised manuscript, please include the following:

-A response letter with a point-by-point reply to each of the referee comments and a description of changes made. Please include the complete referee report in the response letter. Please note that the response letter must be separate to the cover letter to the editors.

-A marked-up version of the manuscript with all changes to the text in a different colored font. Please do not include tracked changes or comments. Please select the file type 'Revised Manuscript - Marked Up' when uploading the manuscript file to our online system.

-A clean version of the manuscript. Please select the file type 'Article File'.

-An updated <https://www.nature.com/documents/nr-editorial-policy-checklist.zip> Editorial Policy checklist, uploaded as a 'Related Manuscript File' type. This checklist is to ensure your paper complies with all relevant editorial policies. If needed, please revise your manuscript in response to these points. Please note that this form is a dynamic 'smart pdf' and must therefore be downloaded and completed in Adobe Reader. Clicking this link will download a zip file containing the pdf.

In the likely event that your manuscript is accepted we will provide detailed guidance on our journal policies and formatting. You may however wish to ensure that the manuscript complies with our house style at this stage. See our style and formatting guide (<https://www.nature.com/documents/commsj-phys-style-formatting-guide-accept.pdf>) and checklist (<https://www.nature.com/documents/commsj-phys-style-formatting-checklist-article.pdf>) for reference.

Data availability statements and data citations policy: All Communications Materials manuscripts must include a section titled "Data Availability" at the end of the Methods section or main text (if no Methods). More information on this policy, and a list of examples, is available at <http://www.nature.com/authors/policies/data/data-availability-statements-data-citations.pdf>.

- Accession codes for deposited data
- Other unique identifiers (such as DOIs and hyperlinks for any other datasets)
- At a minimum, a statement confirming that all relevant data are available from the authors
- If applicable, a statement regarding data available with restrictions
- If a dataset has a Digital Object Identifier (DOI) as its unique identifier, we strongly encourage including this in the Reference list and citing the dataset in the Data Availability Statement.

DATA SOURCES: We strongly encourage authors to deposit all new data associated with the paper in a persistent repository where they can be freely and enduringly accessed. We recommend submitting the data to discipline-specific, community-recognized repositories, where possible and a list of recommended repositories is provided at <http://www.nature.com/sdata/policies/repositories>.

If a community resource is unavailable, data can be submitted to generalist repositories such as [figshare](https://figshare.com/) or [Dryad Digital Repository](http://datadryad.org/). Please provide a unique identifier for the data (for example a DOI or a permanent URL) in the data availability statement, if possible. If the repository does not provide identifiers, we encourage authors to supply the search terms that will return the data. For data that have been obtained from publically available sources, please provide a URL and the specific data product name in the data availability statement. Data with a DOI should be further cited in the methods reference section.

Please use the following link to submit your documents:

Link Redacted

We hope to receive your revised paper within six weeks; please let us know if you aren't able to submit it within this time so that we can discuss how best to proceed. If we don't hear from you, and the revision process takes significantly longer, we will close your file. In this event, we will still be happy to reconsider your paper at a later date, as long as nothing similar has been accepted for publication at Communications Materials or published elsewhere in the meantime.

Please do not hesitate to contact me if you have any questions or would like to discuss these revisions further. We look forward to seeing the revised manuscript and thank you for the opportunity to review your work.

Best regards,

Dr Aldo Isidori
Senior Editor
Communications Materials

Reviewers' comments:

Reviewer #1 (Remarks to the Author):

In this work, C. M. Yim et al., reports on the observation of non-periodic superstructures in the Pd-terminated surface of PdCrO₂ induced by the adsorption of hydrogen atoms. They characterized the structural and electronic properties of the pristine and reconstructed surface through extensive low temperature STM/STS measurements and complemented by ab-initio calculation. Additionally, they performed inelastic tunneling spectroscopy in order to characterize the vibrational modes of hydrogen atoms which serve as a central proof of the presence H in the surfaces.

I find the manuscript very well-written, timely and easy to follow. The experimental data is of a very high-quality standard and compelling, and the interpretation is sound. I do not find any major comments regarding the technical and scientific part of the article. It is a beautiful job. The only weakness is that the topic of this manuscript aligns closer to a fundamental surface science study, since it is quite common that adsorbates of different nature modify the surface structural and electronic properties of a material (for examples in metal or semiconductors). This may raise some concerns regarding its suitability for this Journal, which I would like to leave the final decision to the editorial team. In any case, I find myself accepting this article as it is, without further revision.

Reviewer #2 (Remarks to the Author):

Authors report a scanning tunneling microscopy (STM) study of atomic hydrogen adsorption on PdCrO₂ single crystal. The

PdCrO₂ crystal is cleaved at cryogenic temperatures exposing areas of clean Pd and CrO₂ terminations. On Pd areas, dissociative hydrogen adsorption from the residual vacuum can occur forming a complex adsorption pattern. Authors describe the hydrogen pattern as a combination of six defined recurring tiling structures. A neural network approach is proposed to effectively characterize the concentration of the observed structures. It is proposed that the hydrogen adsorption is driven by interplay between two processes: surface charge compensation of the polar Pd termination and inter-hydrogen repulsion.

I like the manuscript and would like to recommend publishing in Nature's Communications Materials, it's well-written, the presented data is of high quality and supports the claims of the paper. However, I find it that the control over the hydrogen adsorption on the surface is missing. In the image Figure S12, authors show a hydrogen patterns formed after cleaving at 77K and from what I can see the majority of the tiled hydrogen structures are T1 (from nomenclature of the paper). At the very least the concentration between the T1~7 clusters appears to be very different to the 20K cleaved sample. In principle, this shows that the preparation conditions may affect the concentrations of the adsorbed hydrogen clusters. Is it possible to intentionally prepare samples of predominant hydrogen tiled structures on the surface? I think the authors should comment on it.

Communications Materials is committed to improving transparency in authorship. As part of our efforts in this direction, we are now requesting that all authors identified as 'corresponding author' create and link their Open Researcher and Contributor Identifier (ORCID) with their account on the Manuscript Tracking System prior to acceptance. ORCID helps the scientific community achieve unambiguous attribution of all scholarly contributions. You can create and link your ORCID from the home page of the Manuscript Tracking System by clicking on 'Modify my Springer Nature account' and following the instructions in the link below. Please also inform all co-authors that they can add their ORCIDs to their accounts and that they must do so prior to acceptance.

Version 1:

Decision Letter:

Dear Professor Yim,

Your manuscript titled "Adsorbate-induced formation of a surface-polarity-driven non-periodic superstructure" has now been seen again by Reviewer 2, whose comments appear below. In light of their advice I am delighted to say that we are happy, in principle, to publish a suitably revised version in Communications Materials.

We therefore invite you to edit your manuscript to comply with our journal policies and formatting style in order to maximise the accessibility and therefore the impact of your work.

EDITORIAL REQUESTS

* Your manuscript should comply with our policies and format requirements, detailed in our style and formatting guide (<https://www.nature.com/documents/commsj-phys-style-formatting-guide-accept.pdf>).

* Please edit your manuscript according to the editorial requests in the attached table, and outline revisions made in the right hand column. If you have any questions or concerns about any of our requests, please do not hesitate to contact me. It is important that each request be addressed in order to avoid delays in accepting your manuscript. Please upload the completed table with your manuscript files as a Related Manuscript file.

* The editorial requests table also includes a full list of the files that must be provided upon resubmission. Please upload your files according to this table.

OPEN ACCESS

Communications Materials is a fully open access journal. Articles are made freely accessible on publication. For further

information about article processing charges, open access funding, and advice and support from Nature Research, please visit <https://www.nature.com/commsmat/open-access>

Please use the following link to submit your revised files:

Link Redacted

We hope to hear from you within two weeks; please let us know if the process may take longer.

Best regards,

Dr Aldo Isidori
Senior Editor
Communications Materials

REVIEWERS' COMMENTS:

Reviewer #2 (Remarks to the Author):

I thank the authors for addressing my concerns about the hydrogen adsorption differences between the samples cleaved at different temperatures. I have no additional comments and support publishing the article in Nature's Communications Materials.

Reply to Reviewers' Comments (for submission of revised manuscript no.: COMMSMAT-25-0313A)

We thank the reviewers for spending their valuable time reviewing our manuscript and for providing their insightful comments, which we address on a point-by-point basis below. Our answers/replies are in blue color and italic:

Reviewers' comments:

Reviewer #1 (Remarks to the Author):

In this work, C. M. Yim et al., reports on the observation of non-periodic superstructures in the Pd-terminated surface of PdCrO₂ induced by the adsorption of hydrogen atoms. They characterized the structural and electronic properties of the pristine and reconstructed surface through extensive low temperature STM/STS measurements and complemented by ab-initio calculation. Additionally, they performed inelastic tunneling spectroscopy in order to characterize the vibrational modes of hydrogen atoms which serve as a central proof of the presence H in the surfaces.

I find the manuscript very well-written, timely and easy to follow. The experimental data is of a very high-quality standard and compelling, and the interpretation is sound. I do not find any major comments regarding the technical and scientific part of the article. It is a beautiful job. The only weakness is that the topic of this manuscript aligns closer to a fundamental surface science study, since it is quite common that adsorbates of different nature modify the surface structural and electronic properties of a material (for examples in metal or semiconductors). This may raise some concerns regarding its suitability for this Journal, which I would like to leave the final decision to the editorial team. In any case, I find myself accepting this article as it is, without further revision.

We feel very grateful for the reviewer's recognition of our work.

Reviewer #2 (Remarks to the Author):

Authors report a scanning tunneling microscopy (STM) study of atomic hydrogen adsorption on PdCrO₂ single crystal. The PdCrO₂ crystal is cleaved at cryogenic temperatures exposing areas of clean Pd and CrO₂ terminations. On Pd areas, dissociative hydrogen adsorption from the residual vacuum can occur forming a complex adsorption pattern. Authors describe the hydrogen pattern as a combination of six defined recurring tiling structures. A neural network approach is proposed to effectively characterize the concentration of the observed structures. It is proposed that the hydrogen adsorption is driven by interplay between two processes: surface charge compensation of the polar Pd termination and inter-hydrogen repulsion.

I like the manuscript and would like to recommend publishing in Nature's Communications Materials, it's well-written, the presented data is of high quality and supports the claims of the

paper. However, **I find it that the control over the hydrogen adsorption on the surface is missing.** In the image Figure S12, authors show a hydrogen pattern formed after cleaving at 77K and from what I can see the majority of the tiled hydrogen structures are T1 (from nomenclature of the paper). At the very least the concentration between the T1~7 clusters appear to be very different to the 20K cleaved sample. In principle, this shows that the preparation conditions may affect the concentrations of the adsorbed hydrogen clusters. Is it possible to intentionally prepare samples of predominant hydrogen tiled structures on the surface? I think the authors should comment on it.

We feel very grateful for the reviewer's recognition of our work and appreciate his/her comment on the control over the hydrogen adsorption, which we will address very carefully as follows:

First, we would like to point out that the difference in the surface hydrogen coverage (θ_H) between the two samples are indeed small: the 20K-cleaved sample (Fig. S10) has θ_H of ~ 0.63 ML (Fig. 3b), and the 78K-cleaved sample, populated predominantly by T1 clusters on the surface (Fig. 12), has θ_H of ≥ 0.583 ML (Fig. 3b). We note that this is only the coverage of Pd-terminated areas, if one assumes that about half the surface is CrO_2 terminated, the total average coverage is about 0.3ML. The small difference in θ_H between the two samples suggests that for both, the Pd-terminated areas were saturated by hydrogen from residual vacuum immediately after cleaving, and their resultant cluster distributions on the surface very likely result from different cleaving temperatures and hence different formation kinetics. Even though the coverages of the two samples are different, the resulting tiling structures and their properties (i.e., increase in surface work function, low energy electronic localization and sharp inelastic excitations etc.) are practically identical.

While it would be possible to first prepare clean Pd-terminated samples and then dose the samples with increasing amount of hydrogen and monitor the structural evolution of the adsorbed hydrogen on the sample in STM, it would be very difficult to control the hydrogen coverage on the surface because of the high reactivity between the Pd-terminated surface and hydrogen, as well as the high hydrogen background pressure that naturally exists in any UHV system. Furthermore, it would be even more difficult to obtain the tiling structure of desired cluster distribution due to its high sensitivity to hydrogen coverage.

As our response to the reviewer's comment, we have added to following sentence to the revised manuscript:

"Furthermore, in addition to confirming the composition of the tiling structure depends on the temperature at which they form, we also note that the tiling structure is absent from the 12 K-cleaved sample, which itself reveals a metallic surface (see Fig. S12) with similar spectroscopic signatures as seen on the Pd-terminated surface of PdCoO_2 [38]. We therefore attribute the absence of the tiling on the 12 K-cleaved sample to the huge reduction of the hydrogen partial pressure inside the vacuum chamber as a result of cold trapping by the liquid helium-cooled surfaces during sample cleavage, in turn leading to no adsorption of hydrogen on the sample surface." – page 9